# Post-MVC Cervical Kyphosis Deformity Reduction Using Chiropractic BioPhysics Protocols: 1-Year Follow-Up Case Report

**DOI:** 10.3390/healthcare13192459

**Published:** 2025-09-28

**Authors:** Nicholas J. Smith, Thomas J. Woodham, Miles O. Fortner

**Affiliations:** Western Plains Chiropractic, Gillette, WY 82718, USA; tjwoodham12@gmail.com (T.J.W.); mofortner@gmail.com (M.O.F.)

**Keywords:** cervical, spinal deformity, sagittal balance, degeneration, case report, spinal rehabilitation

## Abstract

**Background/Objectives:** This case represents the successful treatment of cervical spine injury from high-speed rear-impact motor vehicle collision and abnormal cervical kyphosis with left arm radiculopathy, utilizing conservative spine care rehabilitation methods. This patient was treated with a multimodal treatment approach integrating a cervical spine extension traction protocol. **Subject and Methods:** A 50-year-old male with a history of motor vehicle collision presented with left arm radiculopathy, as well as cervical and upper thoracic spine pain. Notably the cervical spine presented with kyphotic deformity. The patient presented, after a being struck during a rear-end motor vehicle collision, with neck, upper back, and left arm radiculopathy. Prescription medication and traditional chiropractic care proved ineffective for substantive symptom and quality-of-life improvement. Treatment frequency was three times per week for eight weeks using the Chiropractic Biophysics^®^ protocol of mirror image (MI^®^) postural exercise, spinal adjustment, and cervical spinal traction. On completion of in-office care, the patient was treated monthly, performed home care at least three times per week, and was re-examined at one year. **Results:** Final examination after eight weeks of care showed significant improvement in cervical lordosis (21.8 degrees), resulting in reduced cervical kyphosis. The patient completed outcome indices before, during, and 12 months after cessation of active care, all indicating improvement. **Conclusions:** This case report demonstrates both subjective and objective improvement in cervical spine kyphosis and attendant symptoms. The successful treatment of chronic pain, peripheral weakness, and radiculopathy with long-term follow-up using CBP care is documented as well. The treatment was designed to improve sagittal balance and reduce radiographic abnormalities evincing spinal misalignment. Administration of subjective, objective, and health-related quality-of-life outcome indices during, following, and 12 months post-treatment are suggestive of long-term efficacy of Chiropractic BioPhysics^®^ (CBP) treatment methods. Larger studies are needed to substantiate this given the limitations of a case report.

## 1. Introduction

Cervical kyphosis is a debilitating deformity linked to a myriad of symptoms and conditions, including neck pain, radiculopathy, myelopathy, reduced range of motion, and more. Loss of anterior disk height secondary to trauma leads to wedging and a further increase in loading of the anterior column, facilitating the progression of kyphosis. Trauma from car crashes causes “snap-through buckling” of the cervical spine due to combined forces through a phenomenon known as “ramping”. This is a term used to describe the upward displacement of the thorax after impact. Ramping of the torso combines with extension and compression of the cervical spine, resulting in the previously mentioned snap-through buckling.

Normal function of the anterior and posterior cervical spine elements maintaining lordosis manage stress on the vertebral endplates and compressive loads. On the contrary, cervical spine kyphosis alters normal mechanical and neurologic function. At the same time, cervical spine kyphosis also has a negative impact on the ability to maintain normal horizontal gaze. The etiology of this condition derives from several possible causes, including trauma, degeneration, infection, ankylosing spondylitis, and iatrogenic post-laminectomy [1].

Importantly, this case also addresses forward head posture (FHP), also known as anterior head carriage. When the static position of the cranium is anterior to the hip joint, there are a range of negative consequences, including a decrease in static balance, temporomandibular dysfunction, and weakening of respiratory function. Further sequelae include decreased cervical range of motion, increased craniovertebral angle, and pain [2]. A systematic review and meta-analysis conducted by Mahmoud et al. found that neck pain and FHP are significantly correlated in adult patients. Importantly, magnitude of FHP and pain intensity are also significantly correlated [3].

Patients with cervical spine kyphosis, FHP, and radiculopathy are often treated with traditional conservative care methods. These may be utilized unless symptoms fail to improve, which often leads to referral for more invasive procedures ranging from injections to surgery. General-practice physicians treat these cases using anti-inflammatories and analgesic medications, and often refer to specialty providers, often starting with physical therapists. Some patients may be referred to physiatrists and pain management doctors, typically being treated with injection therapy, radio frequency ablation, and pain medication, sometimes narcotics. As a last resort, patients may undergo surgical procedures performed by an orthopedist or neurosurgeon, usually discectomies and/or spine fusions. In some cases, these procedures possibly could have been avoided. This case is an example of the possible utility of CBP^®^ treatment protocols to improve abnormal cervical spine kyphosis, though larger prospective studies are needed to substantiate this.

This case highlights a fundamental engineering and anatomy maxim: structure determines function. This parallel is an important one: alteration of the blueprint is an alteration of the system. Human bone meets the engineering definition of a semi-rigid structure; ergo, it can only deform minimally before the threshold for failure is met. Altering normal cervical lordosis creates abnormal stresses and strains, leading to altered function, accelerated degeneration, and eventually deformation of the structure. Given enough time under abnormal load, the structure, in this case vertebral bone, can reach the failure threshold. This case shows the efficacy of applying engineering principals to remedy abnormal cervical kyphosis and create long-term structural change for the benefit of patients. This was achieved without surgical installation of spinal hardware, which carries a risk of complications.

## 2. Case Presentation

### 2.1. Patient History and Clinical Findings

A 50-year-old male presented with neck pain and left arm radiculopathy 9 months after being involved in a 35 mph (high speed) motor vehicle collision (MVC). A high-speed MVC is defined as a vehicle collision at 30 miles per hour or greater. Non-steroidal anti-inflammatory treatment as well as 6 months of traditional chiropractic care proved ineffective for substantive symptom and quality-of-life improvement. The patient works as an automotive body repair technician and has worked in this field for around three decades. The job requires prolonged mechanically deleterious positions for the human frame. It also requires performance of repetitive tasks over long periods of time.

His primary complaint is moderate–severe neck pain with radiculopathy into the left arm and forearm that averages 2/10 on the pain scale and is rated at maximum 6/10 (0 = no pain; 10 = worst pain ever). Most of the time his neck pain is rated as tolerable/moderate. This is typical of patients with radiculopathy: he scored 14/100 on the neck disability index (NDI). The NDI is the most widely used and validated tool for self-evaluation to measure patient disability due to neck pain [4]. A score is obtained in the range of 0–100, with a higher score indicating more disability. The secondary complaint reported is mild mid-back pain, rated as 1/10 on average and 6/10 at maximum (Table 1).

Visual assessment of range of motion (ROM) demonstrated restricted cervical extension and left and right lateral flexion. The patient had restricted range of motion at C2, C5, and C6. He also reported tenderness on palpation throughout the cervical spine musculature. Neurological examination revealed hypoesthesia along the left C6 dermatome. Deep tendon reflexes were all within normal limits. Orthopedic tests, including Jackson’s and Maximal Foraminal compression, were both positive for neck pain with symptoms localized on the left side. Cervical distraction relieved symptoms in the neck and left arm.

Computerized postural analysis showed right lateral flexion of the head in the coronal plane (+RzH). Postural distortions in the sagittal plane showed anterior translation of the thorax relative to the pelvis (+TzT) and cervical flexion relative to thorax (+RxH). The patient also showed pelvic rotation of right posterior buttock relative to the feet (−RyP). Analysis was performed utilizing PostureScreen^®^ digital analysis software Version 25. This tool has been studied and shown to have high inter- and intra-examiner reliability with patients properly attired for the exam [5]. Standing postural analysis has also been shown to withstand scrutiny, demonstrating repeatability and reliability [6].

### 2.2. Radiographic Findings

Weight-bearing upright spine radiographs were taken by a licensed physician in compliance with all state and federal regulations. No malignant pathologies requiring referral were found on any radiographs. Conventional radiography is accepted as an important screening tool for detection of infection, tumors, and degenerative pathologies [7]. Biomechanical parameters were evaluated using the computerized digitization tool PostureRay^®^ [8]. Radiographs revealed multiple abnormal findings. Mild wedging was noted at C4 and C5. Moderate cervical disk degeneration was also noted at these same levels, as was moderate degenerative joint disease. These types of degenerative changes, alongside unilateral arm pain with altered sensation and weakness, represent the classic findings of pathophysiology in radiculopathy/neck pain patients [9].

Anterior translation of the head measured 55.2 mm, whereas the normal measurement is in the range of 0–20 mm. This patient’s cervical kyphosis was secondary to his history of MVC. Haas et al. found a decrease in cervical curve post-MVC and formation of cervical kyphosis from the C3 to C5 segments, plus an increase in FHP [10]. This study showed, on average, a loss of 8° of cervical lordosis, with one case showing a large curve loss of 40.7°. Further, this literature review by Haas et al. suggested two plausible ideas: “(1) that either patients presenting with a pre-existing abnormal cervical spine curvature are predisposed to injury of their cervical spine tissues from the MVC event, or (2) that the MVC event causes a mechanical alteration (buckling event) in the alignment of the lordotic cervical alignment, leading to increased risk of injury and future pain and disability.”

The patient also presented with abnormal hypo-kyphosis of the thoracic spine. Loss of thoracic kyphosis is associated with numerous maladies. These include, but are not limited to, thoracic spine pain, decreased pulmonary function, lumbago, abnormal sagittal balance, and thoracolumbar junction kyphosis. The normal measurement of thoracic spine kyphosis is 44°, and the initial patient measurement was 37°.

The measurement of the cervical spine absolute rotation angles (ARA C2–C7) was found to be 17°, compared to the normal measurement of −42°, using the Harrison posterior tangent method (HPTM) (Figure 1) [11]. The kyphotic apex was at C4/5, where the segmental rotation angle (SRA) measured 15.3° compared to the normal value of −8°. This is a 291.2% loss of segmental cervical lordosis. Anterior translation of the head measured 55.2 mm, whereas the normal measurement is 0 mm.

### 2.3. Initial Pain and Disability Assessments

The Quadruple Visual Analog Score (QVAS), a ubiquitous pain rating index, was utilized to measure subjectively rated improvements [12]. Thus patient-rated index is scored in totality on a scale of 0 to 100, where 0/100 indicates no pain and 100/100 indicates the worst imaginable pain. The initial score for cervical spine pain was 10/100, average pain was 20/100, pain at best was 0/100, and pain at worst was 60/100. The composite initial evaluation score totaled 30/100 (Table 1). The initial QVAS for the thoracic spine scored 10/100, average pain scored 10/100, pain at best scored 0/100, and pain at its worst scored 60/100. The composite thoracic spine QVAS for the thoracic spine was 27/100 (Table 1).

The Neck Disability Index (NDI) outcome measure is used to assess patients’ ability to navigate activities of everyday life unhampered by neck pain. The patient scored 14/100, with 0/100 indicating no pain and 100/100 indicating the worst imaginable. Health Status Questionnaire Rand 36 was implemented to qualify well-being and overall general health perception [13,14]. Contained within it are eight categories, with each presumed to be of equal weight. Scores of 100 represent perfect function without hindrance, and 0 represents no function. The initial assessment showed the following scores containing deficits: health perception score of 77/100, mental health score of 84/100, bodily pain scored as 68/100, and energy/fatigue score of 65/100 (Table 1).

## 3. Methods

The patient was treated in-office across a total of 24 visits over 8 weeks using CBP structural spinal rehabilitation protocols at a frequency of three times per week [15]. The treatments consisted of mirror image spinal manipulative therapy (MI^®^, SMT), MI^®^ postural exercises, and MI^®^ three-point-bending spinal traction. Determination of correct protocols for MI^®^ traction, as well as MI^®^ adjustment and exercise, was carried out using the PostureRay^®^ digital assessment tool, version 25. Location of fulcrum and force vectors for MI^®^ exercise and MI^®^ traction were determined likewise. The patient was started with three minutes of MI^®^ traction, which was increased as tolerated, culminating in a maximum duration of 15 min. Along with abnormal cervical kyphosis, the patient also had a significant loss of thoracic kyphosis, and the seated traction was designed to also take this into account to maximize cervical and thoracic improvement [16]. The primary objective of this traction was to reduce/eliminate cervical kyphosis progressively into lordotic alignment via loading of the ligamentous structures (Figure 2). This is a practical application of viscoelastic creep deformation for long-term spinal alignment improvement [17,18]. The secondary objective was to unload the central and peripheral nervous system for a reduction in adverse mechanical tension and resolution of left arm radicular symptoms [19].

The patient also performed postural strengthening exercises under whole-body vibration of the PowerPlate^®^ (Power Plate Performance Health Systems, LLC, Northbrook, IL, USA) while using the Pro-Lordotic™ cervical spine exerciser, to provide spinal muscular resistance, and a fulcrum. Whole-body vibration has been shown to increase metabolic load, theoretically accelerating the results of therapeutic exercise [20]. The patient was prescribed three separate exercises, and the goal was to increase extension in the mid/lower cervical spine, specifically the C4/5 junction (Figure 3).

Exercises were performed with a 5 s hold (minimum) with a 2 s rest per repetition. There was a total of three separate exercises performed. This exercise routine is meant to strengthen spinal muscle groups to support the desired postural changes and functional deficits accumulated through years of postural distortions. Simplicity and reproducibility are emphasized for both compliance and to allow the patient to perform them at home with minimal equipment.

The patient was also instructed to perform home treatment protocols designed to complement in-office treatment methods to reduce cervical kyphosis and FHP. The goal was to ultimately return to normal cervical lordosis [21,22]. A Pro-Lordotic™ neck exerciser device was provided for home use, and the patient was instructed to use it every day, performing a minimum of 50 repetitions. High compliance was reported, though it was not possible to monitor.

No adverse reactions or symptom exacerbations were reported due to the patient performing any MI^®^ exercises, MI^®^SMT, or MI^®^ traction. Patient compliance was high for both in-office and home care. High satisfaction with the treatment was expressed by the patient, as he felt both in-office and at-home care had high value and contributed positively to his outcomes. This patient’s data was published with expressed and written consent.

## 4. Results

The patient was treated 24 times in-office, and each visit included MI^®^ exercises, MI^®^ postural SMT, and MI^®^ traction. Upon completion of the prescribed treatment plan, all subjective and objective patient measures were repeated. These indices showed improvements in symptomology and reduced dysfunction when compared to initial assessments. The patient reported a 90% improvement in neck and upper back pain as well as a 70% improvement in radicular symptoms in the left arm. Further, the patient reported an improvement in the ability to perform activities of daily living (ADLs) as well as an improvement in posture.

Physical re-examination showed additional metrics of improvement. All cervical ranges of motion (ROMs) were pain-free and within normal limits, save for left and right rotation. Orthopedic tests showed improvement relative to Jackosn’s Cervical Compression and Maximal Foraminal Compression test, as these were re-tested as within normal limits. Given the increase in attendant global cervical spine extension measured on radiographs (Figure 4), indicating a decrease in adverse mechanical tension of the central and peripheral nervous system [23], the dermatomal testing results returned to normal limits, as did those of myotome testing.

NDI showed an improvement from the initial scoring of 14/100, decreasing to 0/100 post-treatment. The Rand 36 showed significant improvements in the bodily pain and energy/fatigue categories. Bodily pain initially scored 68/100 and post-treatment scored 90/100. Energy and fatigue initially scored 65/100 and post-treatment scored 80/100. Health perception and mental health scores decreased slightly. The former initially scored 77/100 and finally scored 72/100. The latter initially scored 84/100 and finally scored 80/100.

Cervical spine QVAS showed an improvement in current pain, average pain, and pain-at-its-worst categories. Current pain decreased from 1/10 to 0/10. Average pain also decreased, initially scoring 2/10 and scoring 1/10 post-treatment. Finally, pain at its worst decreased from 6/10 to 5/10. Pain at its best was unchanged, as its score remained at 0/10. The overall composite score improved from 30/100 to 20/100. The thoracic spine QVAS composite score remained unchanged at 27/100. However, categoric changes occurred in all categories, with pain at its best being the exception, as it remained 0/100. Current pain decreased from 1/10 to 0/10, and pain at its worst decreased from 6/10 to 5/10. Interestingly, the patient reported higher average pain than initially: 1/10 to 3/10. This may be attributed to peri-treatment soreness (Table 1).

### 4.1. Post-Treatment Radiographic Outcome Findings

Upon completion of the 24 prescribed treatments, the patient was scheduled for follow-up radiographs four days later. This mandatory gap period ensures the last treatment does not have any abiding influence on the radiographic results. Follow-up radiographs were performed by the same physician as for the initial series. As such, the same instructions were given to the patient to eliminate the influence of the examiner or bias on patient positioning.

Like the initial radiographs, all films were measured digitally using PostureRay^®^ software. Notably the patient achieved improvement relative to cervical lordosis. The cervical curve (C2–7) improved from 17.0° (kyphosis) to −4.8° (global lordosis), a 22° change. This exceeds the findings of Oakley et al. in a 2021 study which reviewed results of CBP^®^ cervical spine rehabilitative cases, which found an average 14° change [24]. Also of interest, the measured forward head posture (FHP) reduced from 55.2 mm to 26.9 mm. This 28.3 mm decrease also exceeded the results of the previously mentioned study, which found an average of 11.9 mm.

While absolute changes were significant, segmental changes showed important and remarkable improvements. The initial radiographic measurement of the segmental rotation of C3/4 (4.3°) was reduced to 1.2°, notably decreasing segmental kyphosis. Even more significantly, C4/5 decreased from 15.3° to 2.7°. Finally, C6/7 increased lordosis from −0.4° to −7.4°, showing numerically significant improvement (Figure 4).

### 4.2. One-Year Follow-Up Findings

After completion of the 24-visit in-office active care phase, the patient was prescribed monthly in-office stabilization care (a total of 13 visits) as well as three–five-times-weekly home care using the CBP^®^ protocol of postural MI^®^ exercise. The patient reported compliance of three times weekly. All assessments, including the physical exam, radiographs, and subjective and objective indices, were performed by the same physician who conducted the initial and final exams in the active-care phase.

The patient’s reported subjective complaints either remained at the improved level of the active-care phase or improved further. The subjectively reported arm, neck, and upper back pain was 100% improved, as was hypoesthesia of the left forearm. A randomized trial by Moustafa et al. demonstrated durability of improvement throughout a 2-year follow-up [25]. In addition, the outcome indices also largely remained constant or improved, with little exception. NDI at the initial exam scored 14 and ameliorated to 0 at the conclusion of in-office care. This number remained at 0 at 13-month follow-up, indicating no return of neck disability. QVAS of the thoracic spine also showed improvement at seven overall points (Table 1).

Rand 36 is a subjectively rated test that allows the patient to rate ten general categories to estimate quality of life, which is another standardized way to assess progress. The categories are health perception; physical functioning; physical, emotional, and social functioning; mental health; bodily pain; and energy/fatigue. The Rand 36 results demonstrate stability of this subject’s improvements, particularly within categories of bodily pain and energy/fatigue. The categories of physical functioning and physical, emotional, and social functioning remained constant at 100/100 throughout the entirety of the subject’s treatment arc within the clinic.

### 4.3. Radiographic Findings

Of interest, the radiographic findings continued to improve to a significant degree, suggesting the importance of patient compliance with home care and stabilization care. This continued improvement reinforces established principles of ligament remodeling alongside principles of viscoelastic creep/relaxation [26,27]. Mensuration of the cervical spine showed the absolute rotation angle of the cervical spine (ARA) or cervical lordosis using the HPTM. Post-treatment ARA measured −4.8°, while re-evaluation at 13 months measured −9.6°, bolstering the value and necessity of monthly maintenance care. Mathematically this patient has transitioned from global cervical kyphosis to global cervical lordosis. This is noteworthy, as cervical kyphosis has technically been described in the literature as a deformity, with this case meeting the criteria for a specific classification, which is beyond the scope of this case report [28].

On a more granular level, segmental values also showed marked improvement in cervical lordosis. C3/4 improved dramatically from 1.2° (kyphosis) to −7.2° (lordosis). C5/6 showed noteworthy change as well: 1.8° to −8.0°. FHP improved as well, measuring 26.9 mm in the previous exam and measuring 22.8 mm at follow-up. Finally, the thoracic spine ARA also showed a 25% improvement in kyphosis, increasing from 37.1° to 46.4°, with 44° being normal (Figure 5).

## 5. Discussion

This case documents a patient presentation of abnormal cervical kyphosis and forward head posture and the negative impacts these conditions have on multiple health factors. Given the frequency of rear-impact MVC-related injuries and this patient’s history of such an event, this case report could be an indicator for the useful application of extension traction protocols. Reduction in cervical kyphosis and FHP is an important component of the musculoskeletal specialism, and more tools and options could benefit patients, as well as the healthcare system at large. Each of these conditions has been shown to have wide-ranging negative impacts, including reduced quality of life, effects on ADLs, reduced range of motion, degenerative joint disease, and radiculopathy, to name a few. This patient showed improvement in the great majority of categories in the PRO, HRQoL, and QVAS; they also showed improvements in biomechanical spine parameters, as measured on radiographs, including cervical lordosis and improved sagittal balance.

The treatment protocols in this case report focus on the area of misalignment through the application of specific traction forces, targeted exercises, and MI^®^ adjustments. Physical therapy can help improve symptoms and function through strength and mobility exercises but cannot predictably address gross misalignments such as cervical kyphosis. It is, however, quite safe, could prove helpful, and is unlikely to harm the patient. On the contrary, the largest prospective multicenter study to date showed that spinal surgeries for adult cervical deformity have a 57% rate of at least one complication post-surgery. Patients showed improvements in the NDI and QOL metrics but at the same time had a reasonably high possibility of experiencing instrumentation failure, dysphasia, and cardiopulmonary events. During the study, 23 of 169 patients died, though this was attributed to the “overall frailty of this patient population” and was labeled “all-cause mortality” [29]. This is not a wholesale discounting of the utility of surgical intervention, as there are patients for whom surgery may be the right choice when all factors are accounted for. CBP^®^ treatment protocols could be a useful spine care approach, though large-scale Randomized Control Trials are needed to substantiate this.

Crucially, digital radiography was used to discover and document the significant deviation in spinal alignment and its consequences relative to joint health. Importantly, low-risk and cost-effective, it is an important portal-of-entry tool for pathology detection. The reliability and repeatability of spine radiography have been well documented. It should be noted that numerous studies have asserted the relative safety of radiographic studies, and there is little evidence to suggest they cause or exacerbate disease processes. Concerns about diseases such as cancer have been based on an outdated linear model of ionizing radiation risk. The dose level of spine radiography is quite low, and there have been no documented cases of cancer attributed to it, thus rendering standard digital radiography a low-risk practice [30].

This case report does have two obvious limitations: its sample size and retrospective design. The most impactful limitation of this case study its sample size. Given this fact, generally applied assumptions cannot be made based on its findings. The included one-year follow-up is valuable, but the long-term status of this subject’s spine and health cannot truly be known at this point. Further, given the occupation of the subject and moderate degenerative joint disease present, it is logical to conclude that stability care may be needed going forward. A future case series could lend credibility to the results of this case report and give weight to the assertion that amelioration of abnormal cervical kyphosis is an important action relative to long-term patient quality of life. CBP^®^ methods with cervical extension traction have been shown to increase cervical lordosis consistently and predictably [31].

## 6. Conclusions

Cervical spine kyphosis is an abnormal adverse finding that, for maximum patient benefit, needs to be addressed regardless of etiology. The consequences of this condition extend beyond simple neck pain; it can reduce quality of life, create progressive disability, and place strain on a wide range of aspects of the patient’s life. This case report suggests the efficacy of CBP^®^ care and its positive impact on QOL, PRO, and disability indices. Further, the combination of cervical spine kyphosis and forward head posture is a potent detriment to patient health and well-being. Their combined long-term sequelae are impactful, as are the benefits of ameliorating them.

## Figures and Tables

**Figure 1 healthcare-13-02459-f001:**
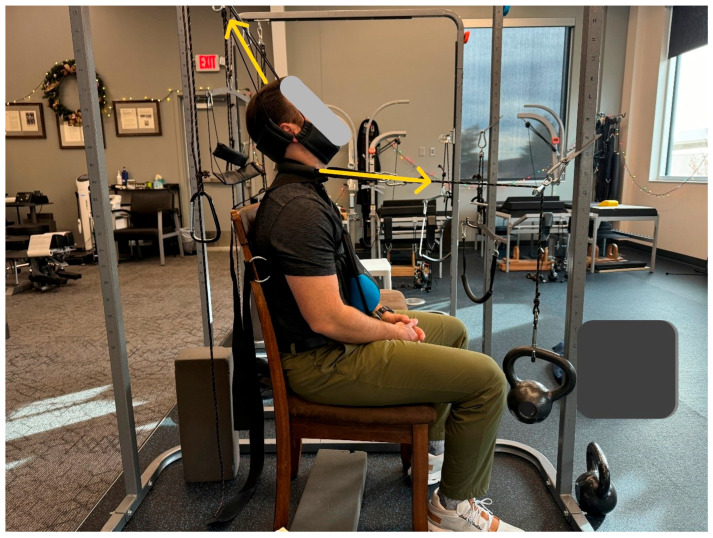
MI^®^ 2-way cervical spine traction setup. The patient is seated with their ribcage fastened to chair at the axilla level and a 10lb weight around the neck to induce thoracic flexion. There are two distinct loads applied: (1) cephalad–posterior to unload the spine while extending and posteriorly translating, and (2) a posterior–anterior load applied mid-cervical spine at an angle 5° below horizontal, with the purpose of this extension load being to reduce kyphosis (flexion) of the mid–lower cervical spine and regain normal cervical lordosis [1]. All loads are applied according to patient tolerance, progressively increasing in time and weight throughout the treatment program. Time: 15 min (maximum).

**Figure 2 healthcare-13-02459-f002:**
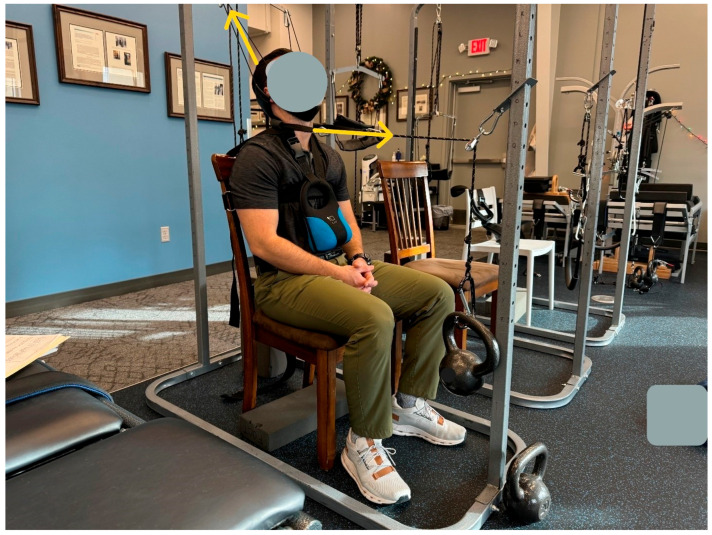
MI^®^ 2-way traction setup and positioning as seen from another angle.

**Figure 3 healthcare-13-02459-f003:**
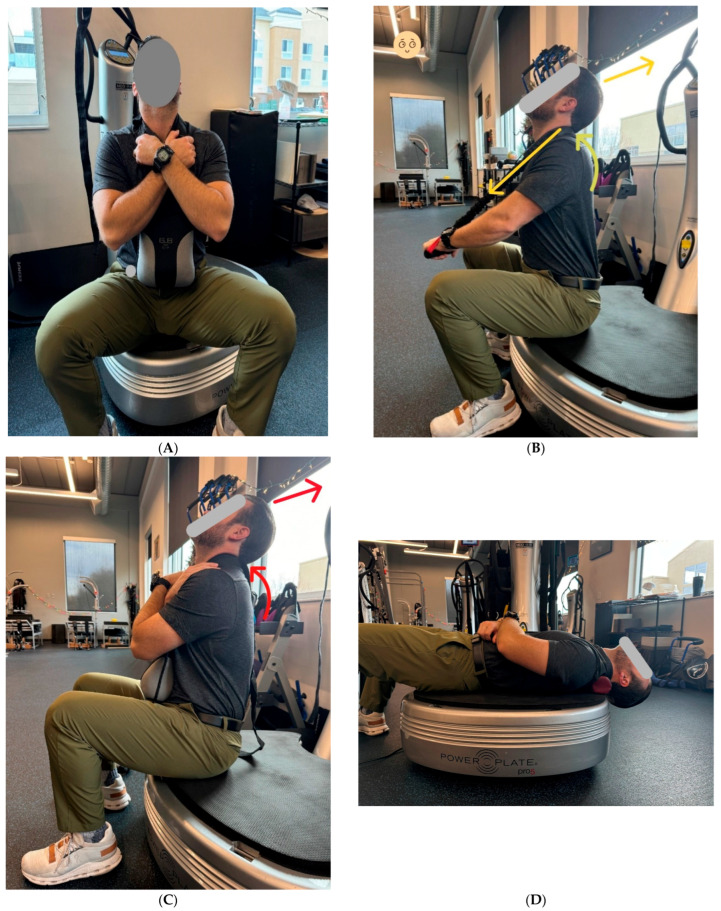
(**A**) The patient is seated on the PowerPlate® (PP) whole-body vibration device. The frequency of vibration creates an involuntary intrinsic muscle contraction by inducing instability. This contraction forces a higher workload of the supporting musculature, creating the physiologic equivalent of higher demand. While under vibration of PP, the patient is instructed to flex the triceps muscles (extend elbow joint) while extending the cervical spine and lifting the chin in both the cephalad and posterior directions. This position is held for five seconds with a two-second rest between repetitions. This exercise is performed for a total of three minutes. (**B**) The patient is also instructed to perform a minimum of 100 repetitions at home every day. The exercise component is an important part of the protocol to retrain the musculature to hold normal posture in place. These exercises, alongside traction, are a critical component of spinal correction. Also, while seated on PP thoracic flexion combined with cervical extension is performed to restore thoracic kyphosis and cervical lordosis. (**C**) While laying supine on the PP, the patient has a foam cylinder positioned under the lower cervical spine. This induces cervical extension and upper thoracic flexion. This final component is a passive modality and does not contain dynamic repetitions, only time: 4 min (**D**).

**Figure 4 healthcare-13-02459-f004:**
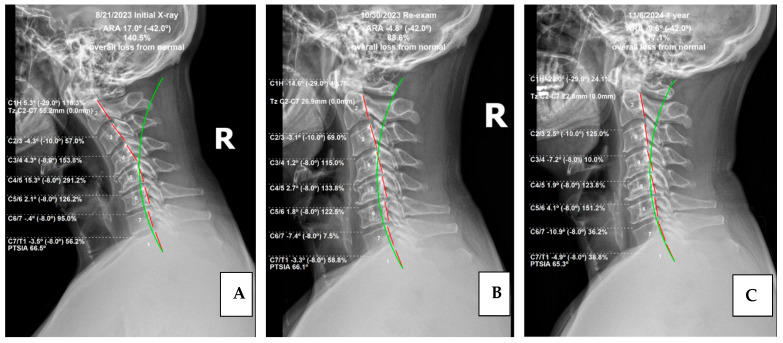
Lateral cervical radiographs. Image (**A**) is the initial radiograph. Image (**B**) is the post-treatment radiograph. Image (**C**) is the one-year follow-up. The broken red line represents the HPTM analysis of the posterior body margin of the individual cervical vertebrae. The green line represents the ideal model of cervical lordosis for comparison of the patient’s measurements relative to normal measurements. These images were measured and digitized using PostureRay^®^ analysis software.

**Figure 5 healthcare-13-02459-f005:**
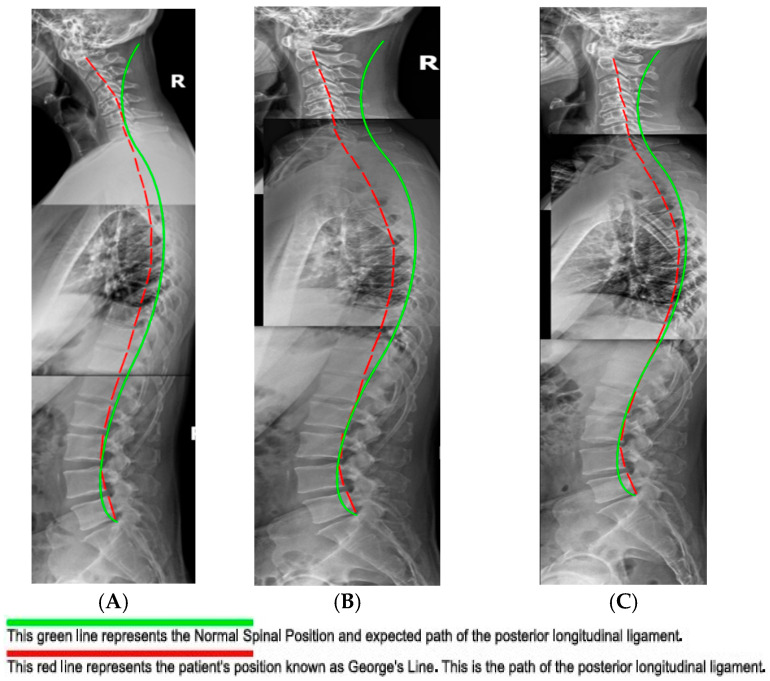
Though this case highlights the importance of addressing abnormal cervical kyphosis, there is another significant radiographic finding: thoracic hypo-kyphosis. (**A**) This patient presented with hypo-kyphosis measuring 36.4° (Normal = 44°). (**B**) The first re-exam showed a minimal increase at 37.1°; and (**C**) at one-year follow-up exam measured a 10° increase, with an improvement to 46.4°. This is an extra benefit for the patient and reinforces the importance of home care and longevity of the CBP^®^ method of spine rehab. The images are, from left to right, the initial exam, re-exam, and one-year follow-up.

**Table 1 healthcare-13-02459-t001:** Patient outcome indices.

Category	28 August 2023	30 October 2023	6 November 2024
**RAND-36**			
Health Perception	77	72	72
Physical Functioning	100	100	100
Physical	100	100	100
Emotional	100	100	100
Social Functioning	100	100	100
Mental Health	84	80	84
Bodily Pain	68	90	90
Energy/Fatigue	65	80	80
**Neck Disability Index**			
NDI Score	14	0	0
**Cervical QVAS**			
Cervical Current Pain	1	0	3
Cervical Average Pain	2	1	3
Cervical Pain at Best	0	0	0
Cervical Worst Pain	6	5	6
Cervical QVAS Total	30	20	30
**Thoracic QVAS**			
Thoracic Current Pain	1	0	0
Thoracic Average Pain	1	3	2
Thoracic Pain at Best	0	0	0
Thoracic Worst Pain	6	5	4
Thoracic QVAS Total	27	27	20
**Current Change**	Health Perception: 0; Physical Functioning: 0; Physical: 0; Emotional: 0; Social Functioning: 0; Mental Health: 4; Bodily Pain: 0; Energy/Fatigue: 0; NDI Score: 0; Cervical Current Pain: 3; Cervical Average Pain: 2; Cervical Pain at Best: 0; Cervical Worst Pain: 1; Cervical QVAS Total: 10; Thoracic Current Pain: 0; Thoracic Average Pain: −1; Thoracic Pain at Best: 0; Thoracic Worst Pain: −1; Thoracic QVAS Total: −7		
**Overall Change**	Health Perception: −5, Physical Functioning: 0, Physical: 0, Emotional: 0, Social Functioning: 0, Mental Health: 0, Bodily Pain: 22, Energy/Fatigue: 15, NDI Score: −14, Cervical Current Pain: 2, Cervical Average Pain: 1, Cervical Pain at Best: 0, Cervical Worst Pain: 0, Cervical QVAS Total: 0, Thoracic Current Pain: −1, Thoracic Average Pain: 1, Thoracic Pain at Best: 0, Thoracic Worst Pain: −2, Thoracic QVAS Total: −7		

## Data Availability

The original contributions presented in this study are included in the article. Further inquiries can be directed to the corresponding author.

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
