# Peer review of "Post-MVC Cervical Kyphosis Deformity Reduction Using Chiropractic BioPhysics Protocols: 1-Year Follow-Up Case Report"

_healthcare, 2025, doi:10.3390/healthcare13192459_

Round 1
Reviewer 1 Report
Comments and Suggestions for Authors
The manuscript presents a single-patient case study evaluating the long-term effectiveness of Chiropractic BioPhysics® (CBP) protocols in reducing cervical kyphosis and forward head posture (FHP) following a motor vehicle collision (MVC). The subject showed both radiographic and subjective improvements after a multimodal CBP-based rehabilitation program.
- The strong claims about long-term efficacy should be clearly toned down and framed as suggestive, not definitive.
- roles must be more transparently addressed in the discussion, especially in relation to potential bias.
- no direct comparison to other modalities or placebo-controlled designs was made.
- tone should be more academic and less promotional, especially in the Discussion and Conclusion sections.
Author Response
To Whom It May Concern,
Your comments and effort of review are appreciated. I've addressed them in the revision process.
- Strident, promotional and firmly assertive language has been removed or changed to be suggestive per your recommendation.
- We have made every effort to remain as objective as possible and felt the case could have value for the future benefit of patients with Adult Cervical Deformity. Initially, as the lead author, I was to be compensated for my time compiling data and leading the writing process on the paper. However, some relationships between that authors have changed, and I have not and will not receive compensation. The work I've put forth will be pro bono and hopefully a valuable contribution to the literature for the benefit of patients.
- The discussion section now has a brief comparison of CBP extension traction methods/physical therapy/surgical intervention for Adult Spinal Deformity.
- Promotional language has been removed.
Discussion section: Comparison
The treatment protocols in this case report focus on the area of misalignment through the application of specific traction forces, targeted exercises, and MI® adjustments. Physical therapy can help symptoms and function through strength and mobility exercises but cannot predictably address gross misalignments such as cervical kyphosis. It is however, quite safe, could prove helpful and is unlikely to harm the patient. On the contrary, the largest prospective multicenter study to date showed spinal surgeries for adult cervical deformity have a 57% rate of at least one complication post-surgery. Patients showed improvement in NDI and QOL metrics while at the same time had a reasonably high possibility of experiencing instrumentation failure, dysphasia and cardiopulmonary events. During the study 23 of 169 patients died, though this was attributed to the “overall frailty of this patient population” and was labeled “all-cause mortality.” [31] This is not a wholesale discount of the utility of surgical intervention as there are patients for whom surgery may be the right choice when all factors are accounted for. CBP® treatment protocols could be a useful spine care approach, though large-scale Randomized Control Trials are needed to substantiate this.
Sincerely,
Nicholas J. Smith
Reviewer 2 Report
Comments and Suggestions for Authors
Dear Authors:
I have had the opportunity to carefully review your manuscript entitled ‘Post-MVC Cervical Kyphosis Deformity Reduction Using Chiropractic BioPhysics Protocols: 1 Year Follow-Up Case Report’. I appreciate the effort made and the intention to contribute evidence on the management of post-traumatic cervical kyphosis using conservative methods.
Below, I will share with you my observations in sections that I hope will help in the improvement of your valuable work.
Abstract:
- I feel that it would be appropriate to adjust the language of the abstract to use a more descriptive and less affirmative tone (‘improvement is noted’ rather than ‘significant improvement’).
- I recommend including in the abstract an explicit mention of the main limitation: the report of a single case, which does not allow us to infer efficacy or generalise results.
Introduction:
The clinical context of cervical kyphosis, its implications and relationship to cervical trauma are adequately described, but the wording tends to be redundant and uncritical, with a strong focus on the CBP® technique.
Gaps in the literature or the specific need to report this case are not explicitly identified, nor are different therapeutic alternatives mentioned. It is recommended to include a critical review of existing approaches, the prevalence of favourable results in the literature and the true novelty that justifies the publication of this case.
Case presentation
- Although the patient, history, clinical, physical examination and radiological findings are described, relevant information on medical and trauma history and previous therapies beyond the general mention of ‘traditional chiropractic care’ is lacking.
- Improve the clarity of descriptions by avoiding technicalities or promotional references to proprietary devices and techniques, using neutral and patient-centred terms.
- Due to the psychosocial component that involves any patient, it would be interesting to clarify whether there was neurological assessment or assessment by other specialists, and whether spontaneous evolution was considered.
Methodology and intervention protocol
- Given that a specific protocol has been developed for an individual, there is a need to know if there are previous studies on the efficacy of the protocol, and how this approach was chosen for the patient.
- In order to mark a casuistry related to the intervention, I consider including the criteria for progression and adaptation of the treatment, as well as possible adverse effects and their monitoring.
- Given that the intervention programme includes activities that the patient must carry out at home, we need to know if adherence to the programme was measured, or even if any guidelines were established to guarantee and control compliance in a strict manner, as set out in the protocol.
Results
- The analysis of results is based on numerical data presented in a descriptive way, but is sometimes interpreted as ‘significant improvement’. Under the circumstances, it is especially recommended to incorporate information on the MCID (minimum clinically important change) of the scales used, citing reference literature for these tools.
- Follow-up is a point to highlight in this study, although the significance is not contextualised and the results are not compared with the usual evolution of the pathology. Since there is no control group, given the characteristics of the study, comparisons with follow-up studies in patients with similar or different approaches could be included. In addition, it would be useful to explain whether there were additional factors during follow-up, for example, changes in activity, new treatments, habits, diet, rest, and even variation in adherence, as mentioned above.
Discussion
- The discussion tends to magnify the importance of the case, with generalisable statements and little self-criticism.
- I consider that the contrast with the international literature (clinical guidelines, systematic reviews, physiotherapy, surgery) is poorly expressed and there is a clear bias towards the defence of the CBP method.
- There is a particular need to expand the limitations section, explicitly discussing the impossibility of statistical inference, potential biases (placebo, regression to the mean, patient expectations), and the lack of control for confounding factors. This added to contextualising the clinical relevance of changes in the scales used, referencing the MCID, and possible comparison in other populations.
- I believe that caution should be exercised in interpretation and avoid extrapolating results to larger populations.
Ethical considerations
- Although it is mentioned that the participant signed an informed consent, I am concerned about the absence in the wording of a clarification of whether the study was reviewed by an institutional ethics committee or whether it was considered exempt (and under what regulations).
Author Response
To Whom It May Concern,
Your input is duly noted and appreciated. This is my first authorship role, and it's been quite a learning experience. I've just completed major revisions to the Case Report with these critiques in mind. Revisiting the paper after a break has allowed me to read the paper with clearer eyes and see it more objectively, at least as much is possible.
Abstract: Strident, promotional language has been removed throughout the entirety of the case report. Here is the new conclusion of this section: Administration of both subjective, objective and health related quality of life outcome indices during, following and 12 months post-treatment are suggestive of long-term efficacy of Chiropractic BioPhysics® (CBP) treatment methods. Larger studies are needed to substantiate this given the limitations of a case report.
Introduction: has been bolstered to state the novelty of long-term X-ray changes. I see what you mean by the promotional feeling of some of the text and have noticed it in statements such as " the need for an increased number of providers to be trained in these methods." This language has been removed, and the revised "Discussion" section now includes a comparison between CBP methods, physical therapy, and surgical intervention for Aduly Cervical Deformity. This input led me to add another reference about a large multi-clinic post surgical outcome paper.
After scrutinizing the report, I've removed all strident and promotional statements about the technique, replacing assertions with modifiers such as "may" or "could".
Further, your critique of the affirmative tone is noted and you are correct. When asserting "significant improvement" I realize that this is a statement without definition. What defines significant? I understand your point of view. The language has been decluttered and redundancies eliminated where detected.
The discussion section now contains a brief comparison to Physical Therapy and surgical interventions for Adult Cervical Deformity.
Case Presentation: The use of terms referencing proprietary equipment is noted. This is still present but has been toned down and isn't meant to be an advertisement. I've tried to be consistent and write in a clinical way though it seems I may be mixing plain-speak with technical terms. Outcome indices have been made into a composite table, albeit a large one. The section containing "traditional chiropractic" has been revised.
Patients with cervical spine kyphosis, FHP, and radiculopathy are often treated with traditional conservative care methods. These may be utilized unless symptoms fail to improve, which often leads to referral for more invasive procedures ranging from injections to surgery. General practice physicians treat these cases using anti-inflammatories, analgesic medications, and often refer to specialty providers, often starting with Physical Therapists. Some patients may be referred to physiatrists and pain management doctors, typically being treated with injection therapy, radio frequency ablation and pain medication, sometimes narcotics. As a last resort, patients may undergo surgical procedures of an orthopedist or neurosurgeon, usually discectomies and/or spine fusions. In some cases, these procedures possibly could have been avoided. This case is an example of the possible utility of CBP® treatment protocols to improve abnormal cervical spine kyphosis though larger prospective studies are needed to substantiate this.
Methodology and Protocol:
Selection of protocol and studies supporting the use of it. During this section there are only 4 references to the efficacy and rationale though more citations may be needed.
The patient was treated in-office a total of 24 visits over 8 weeks using CBP structural spinal rehabilitation protocols at a frequency of three times per week [15]. The treatments consisted mirror image spinal manipulative therapy (MI®, SMT), MI® postural exercises and MI® three point bending spinal traction. Determination of correct protocols for MI® traction, as well as MI® adjustment and exercise was derived using PostureRay® digital assessment tool. Location of fulcrum and force vectors for MI® exercise and MI® traction were determined likewise. The patient was started with three minutes of MI® traction increased as tolerated, culminating in a maximum duration of 15 minutes. Along with abnormal cervical kyphosis, the patient also had a significant loss of thoracic kyphosis and the seated traction was designed to also take this into account to maximize cervical and thoracic improvement [16]. The primary objective of this traction was to reduce/eliminate cervical kyphosis progressively into lordotic alignment via loading of the ligamentous structures. This is a practical application of viscoelastic creep deformation for long term spinal alignment improvement [17-18]. The secondary objective was to unload the central and peripheral nervous system for reduction of adverse mechanical tension and resolution of left arm radicular symptoms [19].
The patient also performed postural strengthening exercises under whole body vibration of the PowerPlate® (Power Plate Performance Health Systems, LLC, Northbrook, IL, USA) while using the Pro-Lordotic™ cervical spine exerciser to provide spinal muscular resistance and a fulcrum. Whole body vibration has been shown to increase metabolic load, theoretically accelerating the results of therapeutic exercise [20]. The patient was prescribed three separate exercises and the goals were to increase extension in the mid/lower cervical spine specifically C4/5 junction.
Exercises were performed 5 seconds hold (minimum) with a 2 second rest per repetition. There was a total of three separate exercises performed. The exercise routine is meant to strengthen spinal muscle groups to support the desired postural changes and functional deficits accumulated through years of postural distortions. Simplicity and reproducibility are emphasized for both compliance and to allow the patient to perform them at home with minimal equipment.
The patient was also instructed to perform home treatment protocols designed to complement in-office treatment methods to reduce cervical kyphosis and FHP. The goal to ultimately return to normal cervical lordosis. [21-22]. A Pro-Lordotic™ neck exerciser device was provided for home use and the patient was instructed to use every day, performing a minimum of 50 repetitions. High compliance was reported though it was not possible to monitor.
As a clinician, I've yet to find a way to ascertain home-care compliance. Though one of the co-authors has developed an app to increase and monitor it.
Results:
Wording such as "significant improvement" has been eliminated as it is not defined and comes across as promotional and superlative. MCID is a valid point of contention and I will examine how I can incorporate it to create a more technical expression of improvement. For now the paper simply contains numerical reporting. Including other comparative studies would add value to the results, though it seems they are not abundant, at least in the context of a specific condition of Adult Cervical Deformity (cervical kyphosis).
In a number of sections in the paper I've now included a more disclaimer-like and less assertion-based expression of the efficacy of the technique. You are right: the paper comes across as biased and strident. Hopefully you'll find the revised version to be much less dogmatic, strident and extrapolatory.
Ethical Considerations:
We consulted the editor of another peer-reviewed journal and were advised that this Case Report did not require IRB approval. I do not know the annotated code regulation though for future reference I will find out.
Your review was by far the most incisive and complete. Thank you for the input. It has helped me compose a better paper. My second one will be far better than this.
Sincerely,
Nicholas J. Smith
Reviewer 3 Report
Comments and Suggestions for Authors
After revising the manuscript entitled " Post-MVC Cervical Kyphosis Deformity Reduction Using Chiropractic BioPhysics Protocols: 1 Year Follow-Up Case Report," I found it interesting. However, several comments were annotated in the attached PDF file. Please consider them in your revision and address them one by one in your reply document
Sincerely

Generally, it's good. However, it might need some improvement
Author Response
To Whom It May Concern,
Thank you for taking your time to review this paper. After reading it with a fresh set of eyes I've made many improvements. The language was decluttered for flow and coherency. Redundancies of expression of numbers and percentages were removed. The tables were made cleared and easier to read. All outcome indices were made into one large composite table. All strident, overly assertive and promotional language was removed as well. The english was lacking and though not perfect, I believe it to be much improved.
Finally, conflict of interest was made clearer. It was our attempt to write this report as objectively as possible to share what we feel is a case that is worth sharing for the benefit of patients suffering from Adult Cervical Deformity. The relationships between the authors has changed. As the lead author I was to be compensated for my personal time put into the paper. Alas, my efforts will be pro bono and I hope to simply contribute meaningfully to the literature.
I think you will find the revised version to be a more friendly and coherent read.
Thank you,
Nicholas J. Smith
Round 2
Reviewer 1 Report
Comments and Suggestions for Authors
The manuscript is suitable for publication in its current form.
Reviewer 2 Report
Comments and Suggestions for Authors
Dear authors:
I have carefully reviewed the revised version of your manuscript and would like to congratulate you on your work. The effort and dedication you have put into addressing the comments made in the first round of review in an appropriate and detailed manner is clearly evident.
The modifications you have made have significantly improved the clarity, coherence, and scientific rigor of the manuscript. I have no further comments to add and consider that, in its current form, the manuscript meets the quality and rigor requirements of the journal.
I appreciate your willingness and the time you have invested in this review process.
Kind regards,